# The Efficacy of Cocoa Polyphenols in the Treatment of Mild Cognitive Impairment: A Retrospective Study

**DOI:** 10.3390/medicina55050156

**Published:** 2019-05-17

**Authors:** Rocco Salvatore Calabrò, Maria Cristina De Cola, Giuseppe Gervasi, Simona Portaro, Antonino Naro, Maria Accorinti, Alfredo Manuli, Angela Marra, Rosaria De Luca, Placido Bramanti

**Affiliations:** 1Behavioral and Robotic Neurorehab Unit, IRCCS Centro Neurolesi “Bonino Pulejo”, 98123 Messina, Italy; cristina.decola@gmail.com (M.C.D.C.); simona.portaro@irccsme.it (S.P.); antonino.naro@irccsme.it (A.N.); maria.accorinti@irccsme.it (M.A.); alfredo.manuli@irccsme.it (A.M.); angela.marra@irccsme.it (A.M.); rosaria.deluca@irccsme.it (R.D.L.); bramanti.dino@irccsme.it (P.B.); 2Hygiene and Preventive Medicine School, Department of Biomedicine and Prevention, University of Rome, Tor Vergata, 00133 Rome, Italy; giuseppe.gervasi79@gmail.com; 3National Center for Disease Prevention and Health Promotion, National Institute of Health, 00133 Rome, Italy

**Keywords:** mild cognitive impairment, nutraceuticals, dementia, neuropsychological assessment

## Abstract

*Background*: Mild cognitive impairment (MCI) is characterized by cognition impairment that does not interfere with the usual activities of daily living. It is considered to be a transitional stage between normal aging and dementia. No treatment is available for MCI. *Methods:* This retrospective cohort study included 55 patients (29 males and 26 females, aged 56–75 years) with a diagnosis of amnestic MCI who attended the Center for Cognitive Disorder and Dementia of the IRCCS Centro Neurolesi Bonino Pulejo (Messina, Italy) between January and December of 2017. As we aimed to evaluate the effect of cocoa polyphenols on cognition, the study population was separated into two groups depending on the change in their Mini-Mental State Examination (MMSE) score at a one-year follow-up. *Results:* Compared to G2 (i.e., patients with a worsening in cognitive functions), the rate of polyphenol intake was significantly higher in patients without a worsening in cognition (i.e., G1) (χ^2^ = 13.79, df = 1, *p*-value < 0.001). By subdividing G1 patients based on whether they improved or were stable at follow-up, we found that 46.2% of those who had improved were treated with polyphenols. *Conclusions*: Dietary supplementation of cocoa flavonoids seems to reduce the progression of MCI to dementia. Further prospective studies with larger sample volumes are required to confirm these promising findings.

## 1. Introduction

Mild cognitive impairment (MCI) is frequent with aging [1]. MCI was first classified as a separate syndrome over 20 years ago. Since then, clinical, imaging, genetic, pathological, and epidemiological studies have been carried out [2]. MCI is considered to be a transitional stage between normal aging and Alzheimer’s disease (AD), and is characterized by cognition impairment that does not interfere with the usual activities of daily living [3,4]. Impairment in memory, attention, and other cognitive domains is more severe in MCI than one would expect for a given age and educational level [5]. MCI prevalence is about 10–20% for people over the age of 65 years, although data contradicting this are available [6,7,8,9].

The rate of MCI-to-dementia conversion depends on the diagnostic criteria employed and the clinical setting (e.g., primary care, specialist clinic, or general population) [10]. MCI prevalence increases with age, and men appear to be at a higher risk [11]. Other risk factors include low educational level, cerebro/cardiovascular diseases (including diabetes and hypertension), vitamin D deficiency, apolipoprotein-E e4 genotype, sleep-disordered breathing, and prior critical illness [12]. A decline in episodic memory is common in AD and amnestic MCI (aMCI) [13].

Neuropsychological and neuroimaging investigations contribute to differentiating between dementia and MCI diagnoses [14,15,16,17,18,19]. No pharmacological or non-pharmacological treatment is currently available for MCI. Different drugs, including donepezil, huperzine-A, vitamin E, and cholinesterase inhibitors, have been tested for their ability to slow or reverse cognitive deterioration. However, there is no converging evidence that these drugs, including cholinesterase inhibitors, can improve neuropsychological test scores or can positively affect the MCI-to-dementia conversion [20,21,22]. Some non-pharmacological interventions, including cognitive training and physical exercise, may be beneficial in this regard [23]. The aim of this study was to retrospectively evaluate the potential role of polyphenols in slowing the MCI-to-dementia conversion.

## 2. Materials and Methods

This retrospective cohort study included patients with a diagnosis of aMCI who attended the Center for Cognitive Disorder and Dementia of the IRCCS Centro Neurolesi Bonino Pulejo (Messina, Italy) between January and December of 2017. Usually, after the diagnosis of aMCI (i.e., at first/second visit) patients are evaluated every six months for at least two years, and are administered the Mini-Mental State Examination (MMSE) to detect cognitive changes. The MMSE measures orientation, immediate and short-term memory, and language functioning, among other things [24]. Specific tests, including the Attentive Matrices (AM) and the Rey Auditory Verbal Learning Test (RAVLT, divided into RAVLI (RAVL immediate) and RAVLR (RAVL recall)) are then administered if deficits in different cognitive domains are found. AM is a neuropsychological tool used to investigate selective attention and visual research and consists of a barrage test of three specific matrices (the number “5” in the first matrices, “2–6” in the second matrices, “1–4–9” in the third matrices). AM is also useful to obtain a better understanding of the interaction between working memory and attentive processes [25]. RAVLT evaluates several functions, including short-term auditory-verbal memory, learning rate and strategies, retroactive and proactive interference, and information retention. It also assesses whether confabulation and confusion in memory processes and differences between learning and retrieval are present. MCI subjects are presented with a list of 15 unrelated words five times. Next, patients are provided with another list of 15 unrelated words, and are then asked to repeat the original 15-word list. The test is then repeated after 30 min [26].

In order to detect possible risk factors or protectors, during their first visit an extensive personal and medical history is obtained and patients verbally report on their habits regarding the consumption of beverages and food containing polyphenols (e.g., coffee, green tea, wine, fruit, and vegetables).

Adherence to the study was evaluated through information collected at a follow-up visit (by searching on the available database).

Patients with MCI can be treated with nutraceuticals, and only a few of them with cholinesterase inhibitors. One of the most commonly used nutraceuticals, Mexenion^®^, is composed of cocoa polyphenols (the main component, 240 mg/sachet), *Bacopa monnieri* (100 mg/sachet), group B vitamins, vitamin E, and folic acid.

To evaluate the role of polyphenols in MCI-to-dementia conversion, we selected patients who had been diagnosed with aMCI within the last year (according to DSM-V criteria) and were treated with Mexenion^®^.

The exclusion criteria were: (i) a diagnosis of dementia; (ii) an age of over 75 years; (iii) the presence of depression and other psychiatric illnesses; (iv) the use of drugs or other nutraceuticals acting on cognitive deficit; (v) the use of antidepressants or other psychoactive drugs.

To evaluate the effect of Mexenion^®^ on cognition, the study population was divided into two groups depending on the change in MMSE score at the one-year follow-up. Thus, the first group (G1) included patients whose MMSE score was stable or increased (i.e., indicating a slowing or reversing of the cognitive decline), whereas the second group (G2) included patients whose MMSE score decreased at least 3 points since the diagnosis.

The Hospital Research Ethical Committee of the IRCCS Neurolesi approved this study (IRCCSME.CE 13/2018), stating that obtaining informed consent was not necessary due to the retrospective design of the study and the fact that all participants were anonymized.

Statistical analysis was performed using Stata 14.1 software, with the level of significance (alpha) set to 0.05. Continuous variable data were expressed as means ± SD, whereas categorical variables were expressed as frequencies and percentages. The χ^2^ test with continuity correction was used to assess the statistical differences in proportions, whereas the unpaired student’s *t*-test was used to compare continuous variables. To assess the association between the consumption of Mexenion^®^ and the slowing or reversing of cognitive decline, we used the Fisher’s exact test, because our sample size was small, and we reported odds ratios (OR) and the associated 95% confidence intervals (CI). We also performed a multiple logistic regression to adjust for patient characteristics (e.g., sex, age, and education) by using the chi-square goodness of fit test and used the Receiver Operating Characteristic (ROC) curve as a post hoc test.

## 3. Results

Of the 109 patients initially screened, 54 were excluded (40 because they were receiving treatment with other nutraceuticals and 14 due to lack of follow-up records of cognitive evaluation). Finally, 55 patients (29 males and 26 females) aged 56 to 75 years were included in this study.

The clinical-demographical characteristics of the sample are reported in Table 1. Compared to G2 (i.e., patients with a worsening in cognitive functions), the rate of Mexenion® intake was significantly higher in patients without a worsening in cognition (G1) (χ^2^ = 13.79, df = 1, *p*-value < 0.001). Indeed, the average change in MMSE score after one year was 1.77 ± 1.68 in G1 and −4.19 ± 1.87 in G2, which is a statistically significant difference (t = −11.05, df = 25.5, *p* < 0.001). Similarly, we observed significant differences between the score changes of patients in G1 and G2 for AM (which after one year was 2.23 ± 5.10 in G1 and −4.69 ± 4.45 in G2, *p* < 0.001) and RAVLT (*p* < 0.001 for RAVLI and *p* = 0.003 for RAVLR). Moreover, the two groups were homogenous in their dietary intake of coffee and wine (about 2–3 cups of coffee per day and 1 glass of wine during meals), as well as in their daily portions of fruit and vegetables. A significant difference was found only in green tea intake: patients in G2 consumed more green tea per day than those in G1 (*p* = 0.01).

By subdividing G1 patients based on whether they improved or were stable at the follow-up, we found that 46.2% of those who improved were treated with Mexenion^®^, whereas about half of the remaining 53.8% were treated with the nutraceutical and were stable (χ^2^ = 0.28, df = 1, *p*-value = 0.59). Moreover, 44% of patients who did not take Mexenion^®^ were stable and 56% were worse at follow-up. The results of Fisher’s exact test showed that Mexenion^®^ is a potential protective factor for slowing or reversing cognitive decline (*p*-value < 0.001) (Table 2). Indeed, the odds of slowing or reversing cognitive decline were 0.79, CI = (0.36, 1.73) for those without any treatment and 14, CI = (3.33, 58.77) for those taking Mexenion^®^. Because education did not add information to the logistic model, we decided to adjust only for demographic variables (e.g., sex and gender). The results of the logistic model showed that Mexenion^®^ intake was the only variable having a significant protective effect on the outcome, and the Mantel-Haenszel OR increased from 16.44, CI = (2.14, 126.06), to 17.43, CI = (3.33, 91.07), when controlled by demographical characteristics. The χ^2^ goodness of fit test did not reject the hypothesis of correctness of our model (χ^2^ = 35.41, df = 32, *p*-value = 0.31). Finally, the area under the ROC curve was 0.85, indicating that the logistic model was well fitted (Figure 1).

## 4. Discussion

MCI is a well-defined clinical condition that is considered to be a prodromal stage of AD. Therefore, although the presence of a spontaneous recovery has been recently described in about 20% of cases, the good management of subjects with MCI could reduce its progression to dementia [27]. 

The results of our study led us to reject the hypothesis of casualty between nutraceutical intake and the reduction of disease progression. Indeed, 87.5% of the patients with a worsening in cognitive functions did not take Mexenion^®^, whereas, on the contrary, all patients showing an improvement at follow-up had been treated.

No pharmacological treatment is officially available for MCI because the physiopathological basis of neurodegeneration is not completely understood. However, the presence of neurofibrillary tangles, τ-complexes, and β-amyloid plaques have been identified as hallmarks of degenerative processes in the brain. Thus, three different pathways (the inflammatory cascade, glucose metabolism, and neurotrophins) are considered to be the causative agents of neurodegeneration [28]. 

Many authors agree with the hypothesis that neurodegeneration is caused by a triggering of the inflammatory cascade. In particular, it has been demonstrated that an increased level of pro-inflammatory cytokines could alter the blood–brain barrier integrity, leading to a subsequent transmigration of leukocytes inside the brain. The presence of leukocytes activates the chronic oxidative stress cascade, leading to synaptic and neuronal dysfunction [29,30].

Growing evidence shows reduced sensitivity of insulin receptors in many areas of the brain, especially the hippocampus, cerebellum, and hypothalamus. The reduced activity of these areas seems to be associated with neural loss and the typical memory impairment of neurodegeneration [31,32]. Finally, evidence for an association between neural death and reduced activity of neurotrophins, including nerve growth factor and the brain-derived neurotrophic factor, has been obtained. The former is associated with an impairment in cholinergic transmission in the basal forebrain, whilst the latter is closely related to apoptosis, with both causing a significant deficit in learning and memory processes [33].

From the results of our study we can argue that dietary supplementation with Mexenion^®^ is a promising therapeutic approach to reducing age-related cognitive impairment. Indeed, 33.3% of the patients who consumed the nutraceutical were stable at the one-year follow-up, and 60% saw improvements in their cognitive functions (as per MMSE) with regard to attention and memory. 

Desideri et al. carried out the first dietary intervention study on elderly subjects with MCI, demonstrating that the regular intake of cocoa flavonoids might improve cognitive function as well as blood pressure and insulin resistance [34]. We retrospectively confirmed these positive results, as habitual consumers of dietary supplements with antioxidant properties were excluded from the study to avoid confounders (as in the Cocoa, Cognition and Aging Study). However, in our study polyphenols were effective for a selected sample of aMCI patients (with less confounder bias and misdiagnosis rate), and at lower dosage than used by the authors (240 vs. 520 mg). We are not able to state if there is an optimal therapeutic dosage of polyphenols to improve cognitive function. However, the optimal therapeutic dosage may depend on the specific properties of the compounds.

Polyphenols are phytochemicals (found abundantly in natural plant food) with antioxidant properties. More than 8000 polyphenols (mostly flavonoids) have been identified, mainly in tea, wine, chocolates, fruits, vegetables, and extra virgin olive oil.

Even though polyphenols have in vitro antioxidative activity, the in vivo activity could be reduced by the accessibility of the compound to the brain. It is worth noting that dietary polyphenols do not necessarily have the best bioavailability profile due to environmental, food-related, and host-related factors, as well as the polyphenol chemical structure and interactions with other compounds [35,36]. Although there is a lack of convincing evidence on the bioavailability of many polyphenols, growing research is focusing on this issue as food industries concentrate their efforts on so-called functional food.

Polyphenols have multiple potential pharmacological effects on the slowing of neurodegenerative processes. First, the innate scavenger activity of flavonoids is important in the reduction of chronic oxidative stress that leads to neural loss. In addition, some polyphenols, such as catechins, may prevent neurodegeneration by inducing different signaling pathways [37], especially the extracellular-signal-regulated kinase pathway [38]. The latter is involved in the autophagy process, modulation of inner mitochondrial membrane permeability, intracellular glucose transport, releasing cytokines from glia, and regulation of transcription processes mainly related to pro-inflammatory factors and inducible nitric oxide.

All of these molecular actions, which are associated with the concomitant positive effect of polyphenols on endothelial function, are responsible for neuronal plasticity (i.e., increased neuronal spine density, glucose metabolism, and blood perfusion) [39]. Nonetheless, we believe that in addition to the pivotal role of cocoa polyphenols in slowing neurodegeneration, the concomitant positive effects of the other compounds, with regard to *B. monnieri*, should also be taken into consideration [40].

The neuroprotective compounds extracted from *B. monnieri* that could improve cognitive performance include steroidal and triterpenoidal saponins. Few in vivo data are available on the bioavailability, metabolism, and pharmacokinetics of these compounds. Therefore, there is no conclusive evidence for the mechanism of action of *B. monnieri* on cognition, reduction of β-amyloid levels (and of their related neuronal damage), brain stress hormones, neuroinflammation factors, restoration of cholinergic and GABAergic transmission, anti-oxidant and neuroprotection factor release, and the increase of 5-HT levels and cerebral blood flow [40].

Our study presents some limitations. First, a prospective cohort study would allow us to directly use the neuropsychological test scores as the outcome variable, and to enroll more patients in order to reduce the information bias. However, the retrospective design of the study was the most appropriate in the presence of the existing data, and the dichotomization of the MMSE score to select patients should overcome the information bias. Indeed, both the goodness of fit test and the ROC curve indicated a good fitting of our logistic model, which included the main confounders such as age and gender.

Secondly, the study design does not allow for the adequate consideration of factors which evaluate adherence to cocoa consumption (e.g., specific dietary information or body weight), as the prospective study does [35].

We investigated patients affected by aMCI, a subtype of MCI that is more likely to convert into AD. Notably, in the amnestic subtype of MCI some genetic markers, including ApoE, APP, presenilin 1, presenilin 2, and tau protein, may already be present, therefore predicting the possible future development of AD [41]. Other important factors responsible for MCI conversion to AD and AD progression should be taken into account. Indeed, it has been demonstrated that depression favors MCI-to-dementia conversion in small cohorts, but there is insufficient data in large clinical populations [42]. For this reason we excluded patients with psychiatric symptoms, including depression, from the current study. Finally, there is no evidence supporting MMSE as an independently sufficient test to identify people at risk of MCI-to-dementia conversion [43]. Therefore, clinicians require other tests beyond neurophysiological tools in order to better manage these patients [44,45,46].

## 5. Conclusions

In conclusion, dietary supplementation of cocoa flavonoids seems to reduce MCI-to-dementia conversion. Since there is growing evidence of a worldwide association between aging and the risk of dementia, the World Health Organization expects that only good management of dementia risk factors, such as MCI, could reduce the incidence and prevalence of people with dementia.

## Figures and Tables

**Figure 1 medicina-55-00156-f001:**
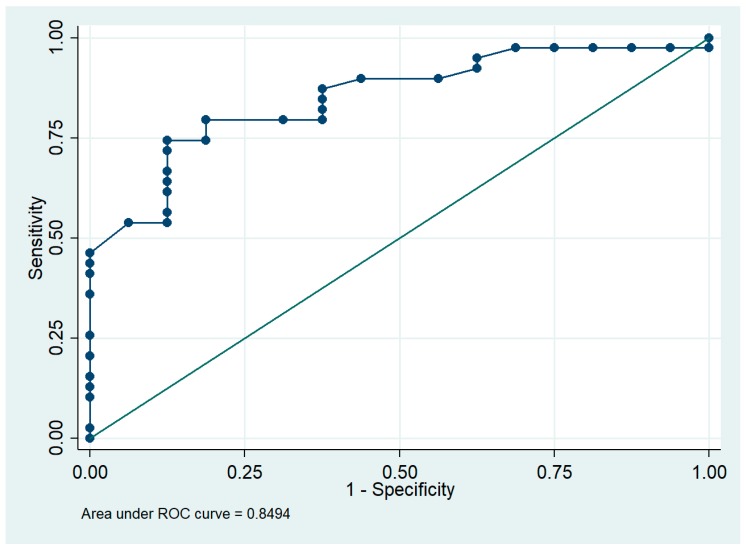
ROC xcurve for the assessment of the logistic model. An area of 1 represents a perfect test and an area of 0.5 represents a worthless test.

**Table 1 medicina-55-00156-t001:** Descriptive analysis of the sample characteristics.

Characteristics	G1 (*n* = 39)	G2 (*n* = 16)	*p*-Value
Sex			
Male	20 (51.28%)	9 (56.25%)	0.97
Female	19 (48.72%)	7 (43.75%)	0.97
Age (years)	66.67 ± 5.27	64.56 ± 4.70	0.16
Education (years)	6.54 ± 2.50	7.12 ± 2.89	0.48
Mexenion^®^ intake	28 (71.79%)	2 (12.50%)	<0.001
Dietary intake			
Coffee	2.63 ± 1.19	2.60 ± 1.22	0.92
Wine	0.97 ± 0.85	1.12 ± 0.83	0.50
Green tea	0.60 ± 0.67	1.16 ± 0.85	0.01
Fruit	1.10 ± 0.80	0.88 ± 0.78	0.31
Vegetables	1.13 ± 0.86	1.12 ± 0.83	0.95

G1 = patients without a worsening in cognitive functions; G2 = patients with a worsening in cognitive functions. *p*-values are based on the χ^2^ test or Student’s *t*-test. Dietary intake is reported as daily cups of beverage or daily portion of food. Continuous variables were expressed as mean ± standard deviation, whereas categorical variables were expressed as frequencies and percentages.

**Table 2 medicina-55-00156-t002:** Risk factors for worsening in cognitive functions at one-year follow-up.

Variables	Odds Ratio	Std. Err.	z-Value	95% Confidence Interval		*p*-value
**Mexenion^®^ consumption**	17.43	14.70	3.39	3.33	91.07	<0.001
**Age**	1.23	0.89	0.28	0.29	5.12	0.20
**Sex**	1.07	0.08	0.97	0.93	1.23	0.88

Pseudo-R^2^ = 0.28; Prob > χ^2^(3) = 0.0003.

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
