# Peer review of "The Efficacy of Cocoa Polyphenols in the Treatment of Mild Cognitive Impairment: A Retrospective Study"

_medicina, 2019, doi:10.3390/medicina55050156_

Round 1
Reviewer 1 Report
The authors replicated the earlier findings of Desideri et al on the cognitive enhancing effect of Cocoa flavonol products in the cohort of MCI subjects, using the retrospective design of classifying the MCI subjects according to the cognitive decline and relate the cognitive changes to the consumption of cocoa. The study can be significant in terms of slowing cognitive decline or even preventing the conversion of MCI to early AD. However, the following issues may have to be discussed in the final text; 1. any explanation of the dosage differences between the current study and early study. 2. perhaps the authors can explain how their methodology has overcome bias; 3. Differences in phyto-chemical properties of the products need to be spelt out : bio-availability. 4. consumption of cocoa data obtained from verbal report or standardized scale of adherence . 5, dietary sources of other polyphenols are unknown and assumed to be balanced among the two cohorts. The dietary intake of coffee and wine have to be comparable in both groups. 6. Perhaps the authors may review the recent studies on polyphenols on conversion of MCI to AD and how to account for non-conversion confounding variable. 7. factors relating to adherence to cocoa consumption have not been mentioned. 8. The exclusion criteria seems to be too stringent: MCI subjects may present as mood changes during the prodromal phase of AD along with subjective memory changes. 9. While the review of the mechanisms of action of polyphenols in AD and MCI is adequate , perhaps the authors may render their opinion as which of the signaling may be more relevant to MCI and conversion from MCI to AD and AD progression.
Author Response
The authors replicated the earlier findings of Desideri et al on the cognitive enhancing effect of Cocoa flavonol products in the cohort of MCI subjects, using the retrospective design of classifying the MCI subjects according to the cognitive decline and relate the cognitive changes to the consumption of cocoa. The study can be significant in terms of slowing cognitive decline or even preventing the conversion of MCI to early AD. However, the following issues may have to be discussed in the final text;
1. any explanation of the dosage differences between the current study and early study.
- As compared to the previous study by Desideri, we are not able to state which is the best intake dosage for polyphenols to exert the positive action on cognition. However, this is maybe related to the properties of the different compounds.
2. perhaps the authors can explain how their methodology has overcome bias;
- The information bias that we mentioned was overcome by the dichotomization of the outcome variable to select the patients, as better explained in the discussion.
3. Differences in phyto-chemical properties of the products need to be spelt out : bio-availability.
-The main differences in phyto-chemical properties of the products has been added, besides the mechanism of action of the other main compound of Mexenion, i.e. bacopa.
4. consumption of cocoa data obtained from verbal report or standardized scale of adherence .
-Consumption of cocoa, such as of food and beverages containing polyphenols, were collected at the first neurological examination by a routine medical interview to detect possible risk factors, as reported within the methods section.
5, dietary sources of other polyphenols are unknown and assumed to be balanced among the two cohorts. The dietary intake of coffee and wine have to be comparable in both groups.
-Dietary intake of food and beverages containing polyphenols was added to our analysis during this revision. We found that the two groups were homogenous in consumption of coffee and wine (about 2-3 cups of coffee per day and 1 glass of wine during meals), as well as in daily portions of fruit and vegetables. They differed only for the green tea, as reported within the results section.
6. Perhaps the authors may review the recent studies on polyphenols on conversion of MCI to AD and how to account for non-conversion confounding variable.
-This issue has been added, as suggested.
7. factors relating to adherence to cocoa consumption have not been mentioned.
-We added this lack as a further study limitation due to its retrospective design, although adherence was retropsectively evaluated at follow up visit.
8. The exclusion criteria seems to be too stringent: MCI subjects may present as mood changes during the prodromal phase of AD along with subjective memory changes.
-The reviewer is right with this concern, that has been added in discussion. However, MCI diagnosis was posed according to DSM-V, and patients with Depressive symptoms were excluded, as per exclusion criteria
9. While the review of the mechanisms of action of polyphenols in AD and MCI is adequate , perhaps the authors may render their opinion as which of the signaling may be more relevant to MCI and conversion from MCI to AD and AD progression.
-As suggested, this issue has been addressed, with regard to the important role of depression and genetic predisposition.
Reviewer 2 Report
Comments to the Authors:
Re: medicina-460532 Calabroet al
In this manuscript, the authors have evaluated the effect of cocoa polyphenols on cognition of a cohort of MCI subjects (29 males and 26 females, aged 56 -75 years), using MMSE score at one-year follow-up. The experimental design is sound and straight-forward, and the experimental methodology is solid and well described. The results are convincing and the findings are interesting as polyphenols in cocoa or green tea are well known neuroprotectants. To this reviewer, the manuscript can be published at the current form.
Author Response
We thank the reviewer for the positive evaluation of our study.